# HAT1: Landscape of Biological Function and Role in Cancer

**DOI:** 10.3390/cells12071075

**Published:** 2023-04-02

**Authors:** Vincenza Capone, Laura Della Torre, Daniela Carannante, Mehrad Babaei, Lucia Altucci, Rosaria Benedetti, Vincenzo Carafa

**Affiliations:** 1Department of Precision Medicine, University of Campania “Luigi Vanvitelli”, Vico De Crecchio 7, 80138 Naples, Italy; vincenza.capone@unicampania.it (V.C.); laura.dellatorre@unicampania.it (L.D.T.); daniela.carannante@unicampania.it (D.C.); mehrad.babaei@unicampania.it (M.B.); rosaria.benedetti@unicampania.it (R.B.); 2Biogem, Molecular Biology and Genetics Research Institute, 83031 Ariano Irpino, Italy; 3IEOS CNR, 80138 Napoli, Italy

**Keywords:** epigenetics, acetylation, HAT1, cancer, inflammation

## Abstract

Histone modifications, as key chromatin regulators, play a pivotal role in the pathogenesis of several diseases, such as cancer. Acetylation, and more specifically lysine acetylation, is a reversible epigenetic process with a fundamental role in cell life, able to target histone and non-histone proteins. This epigenetic modification regulates transcriptional processes and protein activity, stability, and localization. Several studies highlight a specific role for HAT1 in regulating molecular pathways, which are altered in several pathologies, among which is cancer. HAT1 is the first histone acetyltransferase discovered; however, to date, its biological characterization is still unclear. In this review, we summarize and update the current knowledge about the biological function of this acetyltransferase, highlighting recent advances of HAT1 in the pathogenesis of cancer.

## 1. Introduction

Epigenetic modifications play a crucial part in a wide variety of physiological and pathological processes. Different post-translational epigenetic modifications have been discovered and characterized, which are able to affect the proteins targeted, whose alteration is responsible for many disorders [1].

Playing a pivotal role for cellular life, acetylation is considered one of the most important modifications among the post-translational modifications [2]. It can occur in lysine, serine, and threonine amino-acid residue. Due to its involvement in regulating transcription, metabolism, cell proliferation and death, chromatin structure, and DNA strand repair, acetylation in lysine represents one of the most studied epigenetic modifications. It is not surprising that abnormal histone and other protein acetylation are observed in many pathologies, such as cancer [3,4].

Acetylation occurs in all organisms and is the result of the action between two different enzymatic classes: deacetylases (HDACs, Sirtuins), which remove an acetyl group forming the ε-amino group of specific lysine residues, and histone acetyltransferases (HATs), which, instead, add an acetyl group [5]. Lysine acetylation was first discovered in histone proteins; it was considered an exclusive modification of histones [6].

In the last decade, in mammalian cells, this modification has also been found in non-histone proteins—existing in every cell compartment including the nucleus, mitochondria, and cytoplasm—and is involved in several biological processes [7,8]. In this respect, acetyltransferase enzymes are called HATs to highlight their action on histone proteins, and lysine acetyltransferases (KATs) for their action also on lysine present on non-histone proteins [9]. Additionally, it has been discovered in eukaryotes and prokaryotes that proteins can be acetylated on lysine residues enzymatically by lysine acetyltransferases (KATs) and non-enzymatically by acetyl-CoA and/or acetyl-phosphate [10,11]. 

Likewise, the removal of the acetyl group is operated by HDAC and lysine deacetylases (KDAC) to distinguish the action occurring on histone and non-histone proteins, respectively. 

Based on subcellular localization, and substrate recognition, HATs are classified into Type A and Type B enzymes [12]. Type A HATs are linked to transcription genes and divided into five families: GCN5-related N-acetyltransferases (GNAT), MYST, protein 300/CREB-binding protein (p300/CBP), nuclear receptor cofactors (NRCF), and basal transcription factor (TF) family. Two Type B HATs exist, specifically, HAT1 and HAT4, identified as cytoplasmic enzymes responsible for deposition-related acetylation of free, newly synthesized histones before their transport into the nucleus [12]. In this review, we report an update of the biological functions of HAT1 and its involvement in carcinogenesis, highlighting its potential role as a biomarker for cancer.

## 2. Discovery and Enzymatic Activities of HAT1

HAT1 (Hat1p) has been discovered and characterized in Saccharomyces cerevisiae in a complex with another protein Hat2p, able to reinforce the interaction between histone H4 and Hat1p, and increasing its enzymatic action [13].

Hat1p and Hat2p proteins acetylate in vitro free histone H4 on lysine residue 12 (H4K12). The deletion of both genes has no effect on cell growth, with no phenotypic changes, most likely for the compensatory activities of other HATs able to modify histone H4 in the same manner as Hatp1 and Hat2p [14] The role of these two proteins on chromatin function is unclear: they seem to be implicated in telomeric silencing, and only mutated H3 is directly associated with compromising telomeric silencing [14]. In particular, a telomeric silencing defect appears evident when specific aminoacidic residues in histone H3 are mutated, especially when one lysine residue (K14) in the histone H3 tail is substituted with arginine [14]. Further investigation also highlighted a role for K12 of the histone H4, in Hat1p-mediated telomeric silencing [14]. A homologue of Hat1p was identified in mammalian cells; in human cells, it is named HAT1 and shares a 55% sequence similarity with that of yeast [15].

In mammalian, two proteins are homologues of Hat2p, the Retinoblastoma-associated proteins 46 and 48 (Rbp46/RBBP7 and Rbp48), able to bind the retinoblastoma (Rb) [15]. Rbp46 and Rbp48 were found to interact with HDAC1 and HDAC2, highlighting that both may also play a role in transcriptional repression [16]. Rbp46 is known as a chaperone protein. The HAT1/Rbp46 complex is highly conserved along a large variety of eukaryotes and is not required for cell proliferation or viability. HAT1, in complex with Rbp46, acetylates histone H4 on lysins 5 and 12 (H4K5 and H4K12) of the H3/H4 heterodimer. The acetylated H3/H4 dimers interact with anti-silencing factor 1 (Asf1), a chaperone protein involved in the nuclear import of histones, via specific proteins called importins/karyopherins [17,18]. Once in the nucleus, chromatin assembly factor 1 (CAF-1) protein, interacting with proliferating cell nuclear antigen (PCNA), transports histones to the replication forks [17,19,20,21]. In a different chromatin assembly process, as a replacement of Asf1, HAT1 could use another histone chaperone called the nuclear autoantigenic sperm protein (NASP). In addition, HAT1 can remain in complex with histones H3/H4 without transferring these dimers to Asf1 or, in other cases, it can directly enter the nucleus binding nascent chromatin after histone deposition (Figure 1) [17].

Based on its crystal structure, the HAT1 protein is composed of three domains: N-terminal (amino acids 23–136), central (amino acids 137–270), and C-terminal (amino acids 271–341) [22] (Figure 2). No structural analogies were found between the N- and C-terminal domains of HAT1 with other acetyl transferases; instead, the central domain is structurally similar to other acetyltransferases of the GCN5 and MYST families [23].

The usage of glutathione S-transferase (GST)-histone fusion proteins for histones H2A, H2B, H3, and H4, purified in Escherichia Coli, displays that HAT1 acetylates free H4 and identifies it as a preferential substrate for its enzymatic activity. In addition, human HAT1 and yeast Hat1p recognize the same consensus sequence GxGKxG on histones H4 and H2A [22].

Acetyl-coenzyme A (acetyl-CoA) is the donor of the acetyl group necessary for acetyltransferase activity. This enzyme may be generated by ATP citrate lyase (ACLY) from the citrate product of mitochondrial metabolism and acyl-CoA synthetase short chain family member 2 (ACSS2) from acetate. In particular, ACSS2 is found in both the cytoplasmic and nucleus compartments and utilizes acetyl CoA from acetate—in the cytoplasm for lipid biosynthesis and in the nucleus for histone acetylation [24]. Furthermore, ACSS2 can lead to rapid gene activation under metabolic stress conditions, due to the ability of histones to reserve acetate [25].

Structurally, the pocket for the substrate histone H4 and the cofactor acetyl-CoA is between the central and C-terminal domains, within close range of Glu-187, Glu276, and Asp-277 of the enzyme [9]. Glu276 has a crucial role in HAT1 activity. Following a mutational approach, only mutant Glu276 displays a strong decrease in kinetic activity, highlighting its role in HAT1 activity by deprotonation of the ε-amino group K12 of the substrate H4. Acetyl-CoA binds HAT1, at the same site in both human and yeast, but with different molecular interaction. The interactions of the amino residues Phe288 and the side chain of Lys284 in human HAT1, with the ribose and adenine rings of acetyl-CoA, are responsible for a stronger binding than the ones observed in yeast, in which the amino acids involve Arg267 and Asp263, respectively [22].

In addition to the acetyltransferase activity, HAT1 is able to modulate other post-transcriptional modifications (PTMs) such as lysine succinylation and methacrylation. Intermediates of mitochondria metabolism, succinyl-CoA and methacryl-CoA, can be deployed as substrates for the newly discovered enzymatic activities of HAT1. 

## 3. HAT1 Biological Activity

Although HAT1 was the first acetyltransferase discovered, its biological functions together with the cell mechanisms in which it is involved are poorly characterized. The localization of HAT1 has been debated based on different and conflicting scientific evidence [12]. Even though HAT1 has been classified as cytoplasmic type B HAT, it has also been located in the nucleus [26,27,28,29]. Due to its pivotal enzymatic activity, HAT1 acetylates newly synthetized histones in the cytoplasm, in different sites such as H4K5, H4K12, H3K9, H3K18, and H3K27 [24]. Furthermore, it has been reported that it can also acetylate non-histone proteins such as CBP and Hbo1 [30].

Recent studies have reported that HAT1 is able to modulate other post-transcriptional modifications (PTMs), identified as succinyltransferase for non-histone and histone proteins; it succinylates histone H3 on K122 and phosphoglycerate mutase 1 (PGAM1) on K99. Particularly, HAT1-mediated succinylation regulates gene expression in cancer, contributing to cancer initiation and progression [31]. Recently, it has been demonstrated that in response to changes in cell metabolism, HAT1 can use an alternative cofactor such as methacryl-coenzyme A (MC-CoA), an intermediate product of valine mitochondria metabolism [32]. Lysine methacrylation (Kmea) is a novel type of histone PTM; HAT1 has been identified as displaying an unusual enzymatic activity as writer of this new epigenetic modification (Figure 3) [33]. 

The role of HAT1 has been studied in different cell models to better characterize its biological function. Recent findings indicate that HAT1 is transiently localized on chromatin, closer to the DNA replication sites, where it plays a significant role in the regulation of replication fork stall [33]. According to additional studies, the absence of HAT1 increases genomic instability, making cells susceptible to DNA double-strand breaks (DSBs) [34]. Lately, the role of HAT has been demonstrated in replication coupled to chromatin assembly [35].

Scientific evidence in mouse embryonic fibroblasts (MEFs) knockout for HAT1, MEF HAT1^−/−^, highlights the change in the expression of different proteins related to chromatin status. In particular, higher expression levels of topoisomerase 2, the enzyme responsible for DNA spatial conformation, were reported showing several errors in the nascent chromatin and the block in the progression of the replication fork in MEF HAT1^−/−^ [35].

Additionally, a strong decrease in three bromodomain (BRD)-containing proteins—Brahma-related gene-1 (Brg1), BRD-containing Protein 3 (Brd3), and BRD adjacent to zinc finger domain protein 1A (Baz1a)—has been also revealed. These proteins recognize acetylated histones regulating gene expression through a wide range of activities. They can operate as scaffolds promoting the formation of protein complexes, which work as transcriptional co-regulators and transcriptional factors [35].

When the expression levels of these proteins are reduced near the replication forks, there is an alteration of chromatin integrity and of the replication forks [35,36].

Further, an impairment of PCNA function has been observed; PCNA normally binds double-stranded DNA and behaves as a sliding clamp for DNA polymerase, but in MEF HAT1^−/−^, it showed a slowdown in DNA replication and fork stalling [33].

As a consequence of replication fork stalling, higher levels of ataxia telangiectasia and Rad3-related protein (ATR), gamma-H2A family member X (γH2AX), and RAD51 are recruited to repair DNA damage. Scientific evidence supports the role of HAT1 as regulator of DNA repair by homologous recombination. Indeed, upon HAT1 depletion, cells are more vulnerable to DNA damage by altering the global chromatin structure, inhibiting cell proliferation and inducing cell death. The enzymatic activity of HAT1 is necessary to facilitate recruitment and interaction with specific repair factors, such as RAD51; by doing so, it promotes DNA repair [31]. For this process, HAT1 cooperates with the histone chaperone, the histone cell cycle regulator (HIRA). HIRA helps HAT1 in enhancing its histone turnover activity at DSB sites [34].

To corroborate the involvement of HAT1 in DNA organization and replication activity, it has been shown that its absence delays cell recovery following replicative stress.

In MEF HAT1^−/−^ and HAT1 ^+/+^, after the addition of hydroxyurea (which can block replication forks), HAT1 ^+/+^ cells are better able to recover from replication stress [33]. HAT1 is necessary for mammalian viability since its deletion causes different issues in biological development. HAT1-KO mice display genome instability with defects in lung and bones development [37]. DNA replication and chromatin assembly are two of many cellular functions that require energy. Nucleosomes remodeling and the activation of specific gene promoters, mediated by HAT1, are necessary for the different enzymatic reactions related to metabolic pathways [38]. The phosphorylation of HAT1, by 5′ adenosine monophosphate-activated protein kinase (AMPK), contributes to the increase in mitochondrial biogenesis and to the control of mitochondrial membrane potential (ΔΨm) through transactivation of different enzymes such as peroxisome proliferator-activated receptor gamma coactivator 1 (PGC-1α); nuclear respiratory factors 1, 2 (NRF1, NRF2); and uncoupling proteins 2 and 3 (UCP2 and UCP3) [38].

HAT1 plays a metabolic sensor function during the histone acetylation step, using acetyl-CoA derived from glucose metabolism [26]. Moreover, HAT1 is a positive regulator of epidermal growth factor (EGF)-dependent proliferation necessary for its enzymatic activity [24]. Only recently, HAT1 has been involved in the regulation of ATP-binding cassette transporter A1 (ABCA1)-mediated cholesterol efflux, leading to cholesterol accumulation and foam cell formation. HAT1 is a target of miR-486, thus regulating the expression of ABCA1, leading to the accumulation of cholesterol, consequent transformation into foam cells, and the development of atherosclerosis [39]. New insights have identified an unconventional role of HAT1 in controlling the inflammatory response mediated by nuclear factor kappa B (NF-κB). Upon Toll-like receptors’ (TLRs) and tumor necrosis factor receptors’ (TNFRs) activation, the calcium/calmodulin-dependent protein kinase II (CaMK2) activates HAT1 by phosphorylation on Ser361; in turn, HAT1 acetylates the transcriptional regulator promyelocytic leukemia zinc finger (PLZF), promoting the recruitment of a repressor complex in which HDAC3 and NF-κB p50 subunit are present. This activated complex is able to suppress NF-κB activity, leading to the reduction of inflammatory cytokines production [40].

In inflammation-associated atherosclerosis, HAT1 correlates with upregulation of NADPH oxidase 5 (NOX5). It has been shown that, in vitro, the exposure of human macrophages to lipopolysaccharide induces histone H3 acetylation on lysines 9 and 27 (H3K27ac, H3K9ac), enhancing HAT1 expression, which in turn, promotes Nox5 gene promoter activity leading to reactive oxygen species overproduction in atherosclerosis [41].

The role of HAT1 has been also investigated in viral infections. Compared to normal human liver chimeric mice, in hepatitis B virus (HBV)-infected human liver chimeric mice, protein levels of HAT1 are increased, highlighting that HAT1 biological function is related to HBV infection [42]. These findings have also been demonstrated in human cells, following HAT1 depletion. In this latter condition, a reduction in intracellular HBV-DNA levels and an accumulation of covalently closed circular DNA (cccDNA) in HBV-infected human cells was present, resulting in dysregulation of histone H3/H4 assembly on cccDNA. HAT1 has been shown to bind hepatitis B core protein (HBc) via long non-coding RNAs (lncRNA) highly upregulated in liver cancer (HULC) that act as a scaffold in the complex of HAT1/HULC/HBc, which is required for viral replication of HBV cccDNA allowing chromatin assembly by histone acetylation [42].

HAT1 gene activation correlates with monocyte activity in human immunodeficiency virus (HIV) infection. High levels of HAT1 in HIV patients are proportional to the levels of soluble sCD163, which is a sign of monocyte and macrophage activation.

Considering that HAT1 levels were noticeably different between HIV+ patients with sCD163-low and -high levels, HAT1 might be a potential indicator of disease progression. Moreover, HAT1 has been proposed as a molecular target for restoring systemic immunity after viral infections, such as tuberculosis, even if its precise function in the process has still to be clarified [42].

## 4. HAT1 in Cancer

Although the role of type A acetyltransferases in cancer genesis has been widely recognized, less evidence has been reported for type B acetyltransferases. Since the biological function of HAT1 is related to altered molecular mechanisms in human malignancies, a potential role of this acetyltransferase in tumorigenesis has been proposed. Furthermore, a different expression of HAT1 has been highlighted in some types of human cancer, suggesting the controversial role of this protein both as an oncogene and as a tumor suppressor. Compared to normal cells, high levels of HAT1 protein expression were observed in several tumors, such as hepatocellular carcinoma (HCC), esophageal squamous cancer (ESC), uterine leiomyosarcoma (LMS), nasopharyngeal carcinoma (NPC), pancreatic cancer (PC), cervical cancer (CC), lip squamous cell carcinoma (LSCC), actinic cheilitis (AC), breast cancer (BC), colorectal cancer (CRC), diffuse large B-cell lymphoma (DLBCL), and peripheral T-cell lymphoma (PTCL), highlighting its potentially oncogenic role. This overexpression is associated with a poor prognosis and low survival rate. In HCC, high levels of HAT1 are correlated with tumor development and are directly related to the tumor stage [43] (Figure 4).

The overexpression of HAT1 promotes HCC cell proliferation, as a direct effect in the regulation of glucose metabolism [44]. In ESC, HAT1 is overexpressed and associated with poor tumor differentiation [45]. The overexpression is correlated with proliferation, and knockdown of HAT1 leads to an arrest of ESC cells in G2/M as a result of decreased cyclin B1 and D1 expression levels [45]. The oncogenic role of HAT1 is described also in LMS. Immunohistochemical analysis of this rare cancer revealed that HAT1 is expressed more than in leiomyomas, a benign smooth muscle tumor. Hyperthermia treatment for LMS increases the antiproliferative effect of pazopanib, a tyrosine kinase inhibitor [46]. Interestingly, the combined treatment between pazopanib and hyperthermia downregulates HAT1 in LMS cells via circadian locomotor output cycles protein kaput (Clock).

Clock is a transcriptional activator that regulates the circadian rhythms and has intrinsic histone acetyltransferase activity. When Clock expression was suppressed, a substantial decrease in HAT1 transcription and translation was observed. Therefore, the ability of Clock to bind HAT1 to the E-box on the promoter plays a role in the transcriptional control of HAT1 after treatment. Additionally, the effects of pazopanib and hyperthermia were able to downregulate Clock/HAT1 through the RAS-ERK12 pathway. Significant reduction of the acetylation of histone H4K5 and K12 leading to tumor growth inhibition has been reported [46].

In NPC, a correlation between HAT1 and BCL2-like protein-12 (BCL2L12), a member of the BCL2 (B cell-lymphoma 2) family and known as a dominant regulator of programmed cell death, was identified. Both HAT1 and BCL2L12 were found upregulated in NPC and the overexpression of HAT1 was correlated to BCL2L12 transcription. Analysis of NPC tissue has shown high levels of the signal transducer and activator of transcription 5 (STAT5); transcription factor of BCL2L12; and increased levels of H3, H4 acetylation in the promoter region of BCL2L12. Consequently, an increase in antiapoptotic protein occurs through suppression of p53 activity leading to dysregulated apoptosis in cancer cells [47]. Modulating cell growth and immune response, HAT1 promotes tumorigenesis of PC. Recently, a new role for HAT1 as a regulator of programmed death-ligand 1 (PD-L1) at the transcriptional level was identified. The molecular process is mediated by BDR4, which directly binds the promoter region of PD-L1, regulating its transcription. The molecular mechanism is not fully understood, but HAT1-catalyzed acetylation of H4K5 and K12 is required to bind BRD4 to H4 and initiate PD-L1 transcription [44]. Identifying the regulation of PD-L1 in cancer cells may give novel information on the possible development of new drugs for cancer treatment. For example, the discovery of HAT1 inhibitors might be a possible, new way to overcome tumor cells’ immune evasion [48].

Increased levels of acetylation at the K131 residue on the HAT1-mediated chloride intracellular channel-1 (CLIC1) were discovered in CC. Acetylation stabilizes CLIC1, which is not ubiquitinated, leading to tumor development. CLIC1 is frequently overexpressed in tumors and plays a critical role for CC development with a molecular mechanism that involves the activation of NF-κB. The significant function of the CLIC1/NF-κB axis in tumor growth was highlighted by evidence that the pyrrolidine dithiocarbamate, an NF-κB inhibitor, prevents CLIC1-dependent pro-tumor actions [49]. The expression of HAT1 was also evaluated in LSCC and in AC, demonstrating its correlation with two other classes of enzymes, DNA methyltransferases and HDACs. Histopathologic differentiations of LSCC revealed increasing immunopositivity of HAT1, HDAC1, and HDAC2, from well to poorly differentiated carcinoma. Therefore, changes in HAT1 levels occur when the expressions of these enzymes are impaired, leading to less cell differentiation and, consequently, lip carcinogenesis [50].

The specific function of forkhead box P3 (FOXP3)/HAT1 axis is described in BC; it is able to modulate regulatory T-cells (Tregs) infiltration in the tumor microenvironment. The connection was firstly validated between chemokine receptor 4 (CCR4), a tumor-specific chemokine receptor, and FOXP3, shedding light on HAT1′s implication in BC. Particularly, FOXP3 has been discovered to recruit HAT1 on a specific binding site of the CCR4 promoter, inducing changes on the chromatin state from heterochromatin to euchromatin [51].

In CRC, HAT1 cell localization changes significantly among normal tissue, primary cancer, and metastasizing cells; indeed, HAT1 cytoplasmatic localization is increased in primary and metastatic tumors compared to human normal colonic mucosa [40]. Until now, only one evidence has been reported in hematological cancer. In patients with DLBCL and PTCL, HAT1 expression was significantly highly correlated to poorer survival [52].

A contradictory role is observed in LC cells, in which the expression of HAT1 is low; however, this seems related to tumor progression [53].

Few scientific observations support a tumor suppressor role for HAT1. In CRC, the acetylation status of chromatin can be modulated by HAT1 together with sirtuin 7 (SIRT7), allowing centromere protein A (CENP-A) nucleosome assembly and suppressing intestinal tumorigenesis [54]. In the absence of SIRT7, HAT1 resulted as hyperacetylated, leading to the reduction of its acetyltransferase activity, compromising chromatin assembly, and causing aneuploidy status. Moreover, lower histone H4K5/K12 acetylation, due to decreased HAT1 activity, induces dysregulated epithelium regeneration, contributing to a greater incidence of cancer. As a matter of fact, Sirt7 knockout mice show dysregulated epithelial regeneration and higher cancer incidence [54]. The lower expression of HAT1 was observed also in metastatic melanoma, correlating with a resistance to the treatment [55]. In addition, a lower expression was also associated with lung cancer cell (LCC) pathogenesis. HAT1 expression was positively correlated to Fas death receptor expression, which plays an important role in apoptosis. In LCCs, the activation of protease-activated receptor-2 (PAR2) was able to suppress HAT1 and Fas expression differently to normal lung cells, leading to deregulation of the apoptotic process. Interestingly, restoring HAT1 expression increased the levels of FAS, converging to the recovery of apoptotic death in LCC [5] (Table 1).

## 5. Pharmacological Response

HAT1 may also have a role in the pharmacological response to different cancers; in this case, its action may also have a controversial role. One of the current pharmacological treatments approved for HCC is cisplatin, an alkylating agent that induces mutations into DNA and prevents its replication and transcription into RNA directing cancer cells to death. During HCC development, cisplatin resistance often occurs, and it has been demonstrated that HAT1 is involved in this process [44]. HAT1 knockdown in the HCC cells increases the cleaved Poly(ADP-ribose) polymerase (PARP) protein level, restoring cisplatin-induced cell death [44].

In PC cells, HAT1 promotes gemcitabine resistance, identifying plasmacytoma variant translocation 1 (PVT1) as an HAT1 target gene. Acetylating H4 that binds BRD4, HAT1 increases the expression of PVT1 and stimulates its transcription. Conversely, suppressing the expression of BRD4 decreases the expression of PVT1 [56]. The oncogenic role of PVT1 is described for several tumors [57,58,59] and often acts in a complex with the histone–lysine N-methyltransferase enzyme enhancer of zeste homolog 2 (EZH2), conferring resistance to gemcitabine. In PC, both PVT1 and EZH2 have an altered expression. HAT1 can also stabilize the EZH2 protein by binding it into an N-terminal region. This site is specific for the binding of another protein known as ubiquitin protein ligase E3 component N-recognin 4 (UBR4), a ubiquitin ligase, which competes with HAT1 interaction. Upon the suppression of HAT1, UBR4 binds and degrades EZH2, conferring high sensitivity of PC cells to gemcitabine [59].

In castration-resistant prostate cancer (CRPC), the knockdown of HAT1 re-sensitizes PC cells to the action of Enzalutamide action, an androgen-receptor inhibitor [60].

The reduction of HAT1 expression in metastatic melanoma caused the development of the v-raf murine sarcoma viral oncogene homolog B1 (BRAF) inhibitors’ resistance, such as vemurafenib and dabrafenib, kinase inhibitors targeting BRAFV600E in melanoma cells [55].

## 6. Conclusions

Lysine acetylation (Kac) is one of the best characterized post-translational modifications, with a specific role in several cellular functions. In recent years, Kac has been shown to occur on histone and non-histone proteins involved in the regulation of transcription, metabolism, and cell signaling.

Although Type B HATs are not a recent discovery, the molecular mechanisms they regulate and in which they are involved in their pathogenesis are still poorly characterized.

Nevertheless, recent insights have shown the pivotal role of HAT1 in many physiological processes, prompting further investigation aimed to decipher its biological function. Indeed, it has been demonstrated that HAT1 is involved in different cellular processes such as proliferation, DNA replication, cellular metabolism, cell cycle, and cell death.

Furthermore, alterations in HAT1 molecular mechanism correlate with the deregulation of several biological processes, thus promoting cancer onset. Overall, these findings uncover the role of HAT1 as an active player in oncogenic transformation.

Taking into consideration the relevance of HAT1, additional studies are needed to better elucidate its biological function and its involvement in carcinogenesis, identifying it as a possible molecular target for personalized therapy.

## Figures and Tables

**Figure 1 cells-12-01075-f001:**
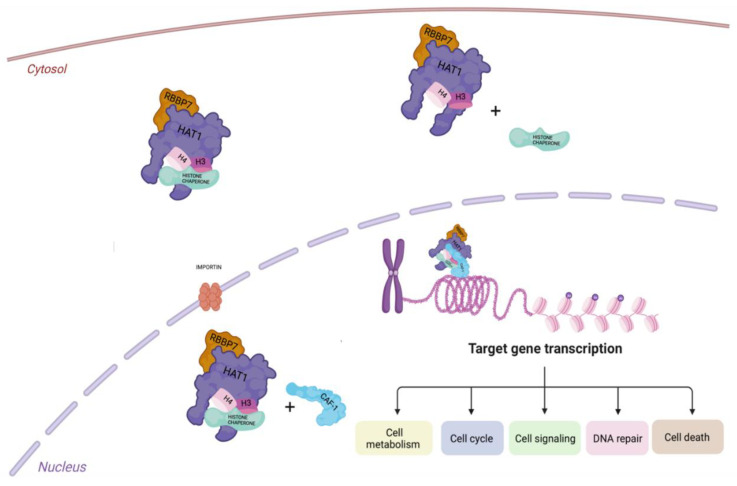
Graphical illustration of HAT1 function. Interaction of the Hat1/Rbap46 complex with histones in the cytoplasm and subsequent transfer into the nucleus via importin, binding histone chaperone protein. Once inside the nucleus, HAT1 interacts with CAF1 and facilitates the pathway for deposition of newly synthesized histones to regulate gene transcription.

**Figure 2 cells-12-01075-f002:**
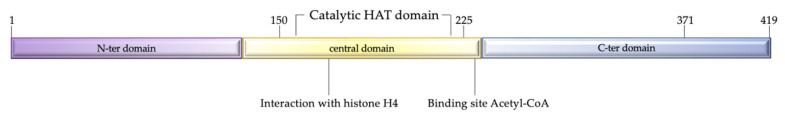
Schematic diagram of the structure of HAT1 protein.

**Figure 3 cells-12-01075-f003:**
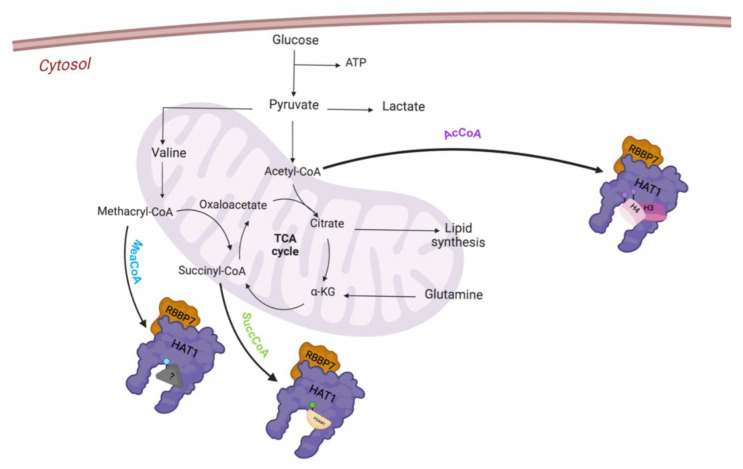
Representative enzymatic function of HAT1.

**Figure 4 cells-12-01075-f004:**
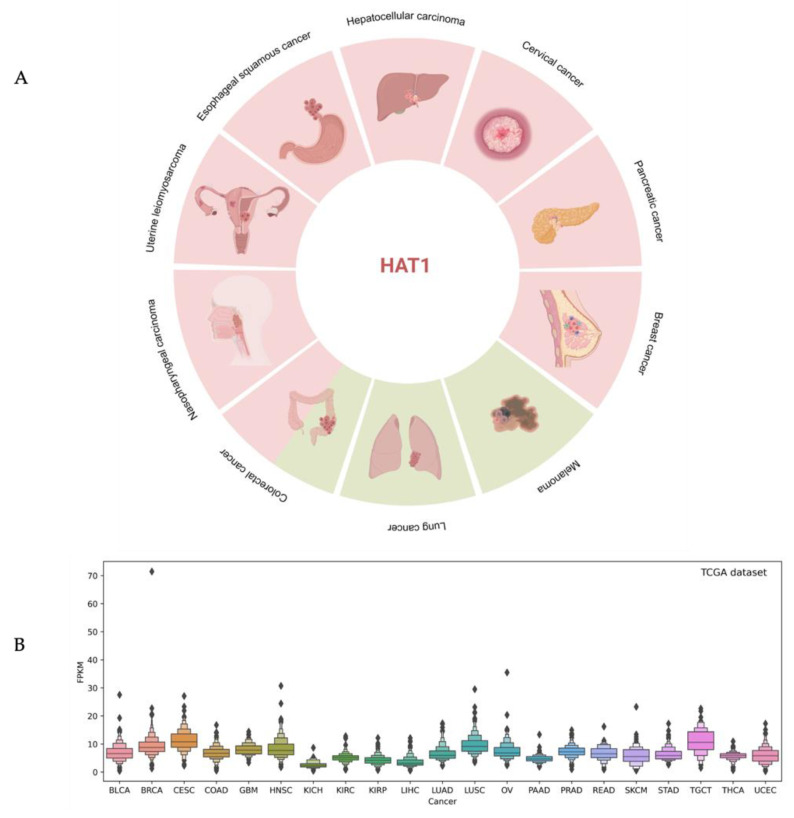
Overview of HAT1 expression in human cancer. (**A**) Schematic representation of the different expression levels of HAT1 in the described cancer. The red and green squares represent the upregulation or downregulation of HAT1 expression levels, respectively. (**B**) The box plot depicts RNA expression levels in 17 different types of cancer from the TCGA dataset; ♦ represent outliers that differ significantly from the rest of the dataset.

**Table 1 cells-12-01075-t001:** Overview of HAT1 expression and related outcomes in different cancer types.

Cancer Type	HAT1 Expression	Outcomes	References
Pancreatic cancer (PC)	High	Resistance to chemotherapy and radiotherapy Poor prognosis	[31]
Colorectal cancer (CRC)	High Low	Not reported	[43,54]
Hepatocellular carcinoma (HCC)	High	Resistance to chemotherapy	[44]
Esophageal squamous cancer (ESC)	High	Poor tumor differentiation	[45]
Uterine leiomyosarcoma (LMS)	High	Poor clinical prognosis Resistant to chemotherapy and radiotherapy	[46]
Nasopharyngeal carcinoma (NPC)	High	Not reported	[47]
Cervical cancer (CC)	High	Poor prognosis	[49]
Lip squamous cell carcinoma (LSCC)	High	Represents 2.1% of all cancers	[50]
Actinic cheilitis (AC)	High	Represents 2.1% of all cancers	[50]
Breast cancer (BC)	High	Poor prognosis	[51]
Diffuse large B-cell lymphoma (DLBCL)	High	Tumor invasion and metastasis Poor prognosis	[52]
Peripheral T-cell lymphoma (PTCL)	High	Tumor invasion and metastasis Poor prognosis	[52]
Lung cancer cell (LCC)	Low	Five-year survival rate is less than 20%	[53]
Melanoma	Low	Resistance to chemotherapy	[55]

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
