# Peer review of "HAT1: Landscape of Biological Function and Role in Cancer"

_cells, 2023, doi:10.3390/cells12071075_

Round 1
Reviewer 1 Report
Overall, this is a well-prepared review, but it would benefit from further clarifications at specific points which will highlighted below. In addition, some of the figures included would benefit from being expanded/revisited to better emphasize the overall narrative of the review. Lastly, the authors should try to highlight on-going questions and point of divergences in the literature related to HAT1, when appropriate, to further the impact of their review during revisions.
- I would rephrase the line 26 “… acetylation is considered the most important modification …” to make it more nuance since phosphorylation and ubiquitylation are more abundant PTMs. I agree that Kac is a very important PTM but stating that it is the most important one seems out of place.
- In the next paragraph (line 33 to 37), the inclusion of non-enzymatic acetylation would be warranted. For a good example, see PMID: 23830618.
- The paragraph discussing the production of acetyl-CoA (line 103 to 108) and figure 2 are a bit misleading. Briefly, two independent pools of acetyl-CoA exist in cells, the mitochondrial and nucleocytoplasmic ones, since acetyl-CoA cannot diffuse or be exported out of mitochondria. Rather, citrate is exported and converted back to acetyl-CoA by ACLY. This is the major metabolic pathway allowing for acetyl-CoA production. As pointed out by the authors, ACSS2 act in a parallel pathway by employing acetate to generate acetyl-CoA in cytoplasm and nucleus. Recent work from the Berger lab, as shown that ACSS2 may be able to employ this activity of ACSS2 to transfer Kac from “reservoir sites” to “active sites” (PMID: 35061542) which may represent the main activity of ACSS2 since acetate is quite low (about 40 uM) in normal plasma. Lastly, there is no discussion of most short-chained acyl-CoA (with the exception of succinylation and methacrylation) in the second section of the review. While these points may go beyond the vision the authors had for their review, the represent areas of intense research in the field that may be of interest to many readers.
- The discussion of Kac readers modules (bromodomains) in lines 155-159 is very limited and could be better included in the review. Readers not familiar with field will not be able to make the link between reduced Kac levels in the HAT1-/- MEFs and reduced recruitment of specific bromodomain containing proteins to replication forks with reduced Kac levels. The interplay of HAT1 with mSWI/SNF complex (BRG1 containing), BET proteins (which includes BRD3) and ISW1 complex (which includes BAZ1A) should also be refined.
- The paragraph structure between line 229 and 241 is quite bizarre (each sentence as a paragraph) and should be reworked.
- Figures 3 and 4 should be revisited and combined if possible. Ideally, the authors would present in a single figure the tumor types that overexpress or downregulate HAT1 expression levels. Additionally, it would be beneficial if the authors further showed in which tumor types the gain or loss of HAT1 expression is predictive of a specific outcomes (survival, disease progression, etc.) since the lines 250-251 are quite broad.
- As previously mentioned, I would appreciate if the overall conclusion highlights key points about HAT1 that should warrant further investigation. As the authors mention, HAT1 is often overshadowed by other KATS (CBP/P300; TIP60; GCN5; etc.) in the literature despite having being identified over 25 years ago.
Author Response
We greatly thank Editor and Referees for their interest in our manuscript entitled “HAT1: landscape of biological function and role in cancer” and for reviewing it. We have carried out the appropriate changes suggested by reviewer implementing the text and figures accordingly.
Below there is a point-by-point response to referee’s questions, reporting our answers in red colour.
We hope that the answers are satisfactory and the improvement of the manuscript makes it acceptable for publication.
Overall, this is a well-prepared review, but it would benefit from further clarifications at specific points which will highlighted below. In addition, some of the figures included would benefit from being expanded/revisited to better emphasize the overall narrative of the review. Lastly, the authors should try to highlight on-going questions and point of divergences in the literature related to HAT1, when appropriate, to further the impact of their review during revisions.
Q1. I would rephrase the line 26 “… acetylation is considered the most important modification …” to make it more nuance since phosphorylation and ubiquitylation are more abundant PTMs. I agree that Kac is a very important PTM but stating that it is the most important one seems out of place.
A1. Thanks for your useful comment. We have changed the text.
Q2. In the next paragraph (line 33 to 37), the inclusion of non-enzymatic acetylation would be warranted. For a good example, see PMID: 23830618.
A2. Thanks for your comment. We have explained the highlighted point better, by adding some information and indicated reference (see lines 43-45, page 1)
Q3. The paragraph discussing the production of acetyl-CoA (line 103 to 108) and figure 2 are a bit misleading. Briefly, two independent pools of acetyl-CoA exist in cells, the mitochondrial and nucleocytoplasmic ones, since acetyl-CoA cannot diffuse or be exported out of mitochondria. Rather, citrate is exported and converted back to acetyl-CoA by ACLY. This is the major metabolic pathway allowing for acetyl-CoA production. As pointed out by the authors, ACSS2 act in a parallel pathway by employing acetate to generate acetyl-CoA in cytoplasm and nucleus. Recent work from the Berger lab, as shown that ACSS2 may be able to employ this activity of ACSS2 to transfer Kac from “reservoir sites” to “active sites” (PMID: 35061542) which may represent the main activity of ACSS2 since acetate is quite low (about 40 uM) in normal plasma. Lastly, there is no discussion of most short-chained acyl-CoA (with the exception of succinylation and methacrylation) in the second section of the review. While these points may go beyond the vision the authors had for their review, the represent areas of intense research in the field that may be of interest to many readers.
A3. Thanks for the useful comment. Following your suggestion, we have better explained the role of the ACSS2 enzyme in the production and use of acetyl-CoA, modifying the text (lines 114-118, page 3) and adding the suggested reference
Q4. The discussion of Kac readers modules (bromodomains) in lines 155-159 is very limited and could be better included in the review. Readers not familiar with field will not be able to make the link between reduced Kac levels in the HAT1-/- MEFs and reduced recruitment of specific bromodomain containing proteins to replication forks with reduced Kac levels. The interplay of HAT1 with mSWI/SNF complex (BRG1 containing), BET proteins (which includes BRD3) and ISW1 complex (which includes BAZ1A) should also be refined.
A4. Thanks for the useful suggestion. We have now explained the function of bromodomain proteins. We have not explained the interaction of HAT1 with the mSWI/SNF complex, because this is well explained for other HATs. No evidences (for what we know) are reported for HAT1. We will expand further the topic, should the reviewer consider it necessary. (see lines 167-170, page 5)
Q5. The paragraph structure between line 229 and 241 is quite bizarre (each sentence as a paragraph) and should be reworked.
A5.Thanks for the comment, we have modified the text. We hope that with the changes made, the sentences will be clearer and more understandable (see page 6, lines 244-248)
Q6. Figures 3 and 4 should be revisited and combined if possible. Ideally, the authors would present in a single figure the tumor types that overexpress or downregulate HAT1 expression levels. Additionally, it would be beneficial if the authors further showed in which tumor types the gain or loss of HAT1 expression is predictive of a specific outcome (survival, disease progression, etc.) since the lines 250-251 are quite broad.
A6. Thanks for the suggestions. We have made a single figure as suggested (now figure 4, page 7). In addition, we have tried to collect the necessary information for cancers described. These information are derived from the papers mentioned, but unfortunately, we were not able to obtain the required information in all cases. Therefore, the table will only contain the ones we found. (Table1, pages 10-11)
Q7. As previously mentioned, I would appreciate if the overall conclusion highlights key points about HAT1 that should warrant further investigation. As the authors mention, HAT1 is often overshadowed by other KATS (CBP/P300; TIP60; GCN5; etc.) in the literature despite having being identified over 25 years ago.
A7. Thanks for the suggestions. We tried to highlight the reason why it is important to investigate HAT1 given its involvement in cancer (lines 406-412, page 11)

Reviewer 2 Report
The authors summarized the biological function of HAT1 and its role in cancer. Since HAT1 is one of the critical epigenetic modifiers, this manuscript which is well organized, will be helpful to many readers in the fields of molecular biology and cancer biology, etc.
However, if the following is supplemented, it could be a better manuscript.
1. It would be better if the structure of the HAT1 was explained along with the schematic diagram.
2. As explained, HAT1 could act as an oncogene and a tumor suppressor gene, which is complicated. Moreover, thus, the explanation and the references should be summarized in a table.
3. The English writing should be extensively proofread. There are some grammatical errors detected.
4. Although epi-modification seems to mean an epigenetic modification, it can be misunderstood as the above modification. So, epimodification should be changed to epigenetic modification.
Author Response
We greatly thank Editor and Referees for their interest in our manuscript entitled “HAT1: landscape of biological function and role in cancer” and for reviewing it. We have carried out the appropriate changes suggested by reviewer implementing the text and figures accordingly.
Below there is a point-by-point response to referee’s questions, reporting our answers in red colour.
We hope that the answers are satisfactory and the improvement of the manuscript makes it acceptable for publication.
The authors summarized the biological function of HAT1 and its role in cancer. Since HAT1 is one of the critical epigenetic modifiers, his manuscript which is well organized, will be helpful to many readers in the fields of molecular biology and cancer biology, etc.
However, if the following is supplemented, it could be a better manuscript.
Q1. It would be better if the structure of the HAT1 was explained along with the schematic diagram.
A1. Thanks for the comment. We have made a schematic diagram as suggested (now Figure 2, page 3)
Q2. As explained, HAT1 could act as an oncogene and a tumor suppressor gene, which is complicated. Moreover, thus, the explanation and the references should be summarized in a table.
A2. Thank you for your comment. We have summarized the information on high and low expression of HAT1 for each tumour in a table (Table 1, pages 10-11).
Q3. The English writing should be extensively proofread. There are some grammatical errors detected.
A3 Thank you for your comment. We revisioned the manuscript and changed some sentences.
Q4. Although epi-modification seems to mean an epigenetic modification, it can be misunderstood as the above modification. So, epimodification should be changed to epigenetic modification.
A4 Thanks for your useful comment. We have changed the text accordingly.
